# Geographical variation and determinants of women unemployment status in Ethiopia; A multilevel and spatial analysis from 2016 Ethiopia Demographic and Health Survey data

Solomon Sisay Mulugeta[1], Shewayiref Geremew Gebremichael[1], Setegn Muche Fenta [1]*, Berhanu Engidaw Getahun[2]

1 Department of Statistics, College of Natural and Computational Sciences, Debre Tabor University, Debre Tabor, Ethiopia, 2 English Language and Literature, Bahir Dar University, Bahir Dar, Amhara, Ethiopia

* setegn14@gmail.com

## Abstract

**Data Availability Statement:** The survey datasets used in this study was based on a publicly available dataset that is freely available online with no

### Background

Unemployment is a major problem in both developed and developing countries. In Ethiopia, women unemployment is particularly high, and this makes it a grave socio-economic concern. The aim of this study is to assess the spatial distribution and identify the determinant factors of women unemployment in Ethiopia.

### Methods

The data used for the study is the Ethiopian Demographic and Health Surveys of 2016. A total of 15683 women are involved in the study. Global Moran's I statistic and Poisson-based purely spatial scan statistics are employed to explore spatial patterns and detect spatial clusters of women unemployment, respectively. To identify factors associated with women unemployment, multilevel logistic regression model is used.

### Results

A spatial analysis showed that there was a major spatial difference in women unemployment in Ethiopia with Global Moran's index value of 0.3 (p<0.001). The spatial distribution of women's unemployment varied significantly across the country. The major areas of unemployment were Afar and Somalia; southwest Tigray; North and west Oromia, and Eastern and southern parts of Amhara. Women with primary level of education(AOR = 0.88, 95%CI: 0.80, 0.98), secondary and above level of education (AOR = 0.71, 95%CI: 0.62, 0.82), women with rich wealth index (AOR = 0.79, 95% CI: 0.70, 0.90), pregnant women (AOR = 1.24, 95% CI: 1.06, 1.5), women with a male household head(AOR = 1.4, 95% CI: 1.28, 1.50), and urban women(AOR = 0.60, 95% CI: 0.50, 0.70) statistically associated with women unemployment.

participant's identity from http://www.dhsprogram.com/data/available-datasets.cfm. Approval was sought from MEASURE DHS/ICF International and permission was granted for this use.

**Funding:** The author(s) received no specific funding for this work.

**Competing interests:** The authors have declared that no competing interests exist.

**Abbreviations:** AIDS, Acquired Immunodeficiency Syndrome; CSA, Central Statistical Agency; EAs, Enumeration Areas; EDHS, Ethiopia Demographic and Health Survey; HIV, Human Immunodeficiency Virus; ICF, International child fund; ILO, International Labor Organization; LFS, Labor force survey; PHC, Population and house census; SNNPR, Southern Nations, Nationalities, and People's Region; UN, United Nation; UNICEF, United Nations Children's Fund; USAID, United States Agency for International Development; WB, World Bank.

## Conclusion

The unemployment rate of women in Ethiopia showed variation across different clusters. Improving entrepreneurship and women's education, sharing business experiences, supporting entrepreneurs are potential tools for reducing the unemployment women. Moreover, creating community-based programs that prioritize participation of poor households and rural women as well as improving their access to mass media and the labor market is crucial.

## Introduction

Unemployment is a critical issue the world is facing. Nonetheless, the effect and severity of unemployment varies between men and women. Globally, the rate of unemployment is higher among women than it is among men. The quality of employment is also disparate among men and women [1–3].Women, for instance, account to 38.8 percent of all labor force participants. However, women rather than spend more time on unpaid work such as childcare and housework. Compared to the 41 million (1.5 percent) of men, 606 million (21.7 percent) women offer full-time unpaid care globally, unpaid labor takes women 4 hours and 22 minutes per day, while for males, it takes only 2 hours and 15 minutes [4,5].

Unemployment not only causes economic uncertainty but also numerous social problems such as violence, human capital erosion, poverty and civil conflict [6]. It further causes social problems such as desperation, anger, violence and the eventual drift of unemployed women into all forms of criminal behavior [7–9]. High women unemployment cause HIV/AIDS spread in developing countries [10]. Long-term unemployment contributes more to financial deprivation, hunger, insecurity, violence, discontent, family conflict, social loneliness, lack of trust and self-esteem which eventually results in the deterioration of a stable community [10]. Moreover, unemployment is a waste of scarce resources since it results in a loss of potential national output [11,12].

Although Ethiopia is among the fast-growing countries in Africa, it has not been able to make effective use of the workforce necessary for sustaining economic development. The government didn't create adequate job opportunities that accommodate future labor force through adopting a viable workforce policy. Consequently, women in Ethiopia not only lack the economic opportunities that allow them engage in alternative income-generating activities but they also lack alternative income sources that left them dependent on their spouse, and non participant party in decision-making of their household [3,13]. As a result of the numerous barriers they face in different aspects of their life, many Ethiopian women migrate to Arab and European countries [14,15]. The migration causes them several physical and mental health disorders. Various reports indicate that migrants are victims of fraud, forced labor, physical, sexual and psychological harassment by their employers, or smugglers. Many among the respondents experience psychological barrier [16,17].

Unemployment is hence a critical and pressing agenda for the Ethiopian Government. Although the Government commits itself to achieving the objectives of reducing unemployment, no significant change is noticed with respect to the reducing unemployment, and the aggregate job growth performance has remained slow.

Prior studies investigated the determinants of women's unemployment in small urban areas [1,10,18], though only on individual-level factors [1,10,18]. At the community-level, unemployment among women is affected by factors, such as residence, media exposure, region

and cluster (enumeration area) [19,20]. In addition, recent studies that employed conventional logistic regression that disregarded clustering effects investigated the rate of women's unemployment and predicted that women's individual characteristics associated with their unemployment. Yet, observations within a cluster tend to be more similar to observations, and an analysis that neglected this remains inadequate. Disregarding clustering in analysis may lead to overstating or understating the accuracy of the results, and determinant variables may be incorrectly reported as important. Thus, considering the shortcomings of existing studies, this study identified factors associated with women's unemployment using a multi-level logistic regression model that considers the association between responses of interest to respondents from within the same cluster [21,22]. Spatial analyses helped locate hotspots and information to decision makers on strategic planning. The aim of this study is, therefore, to assess the spatial distribution and identify the determinant factors of women unemployment by adopting a multilevel model.

## Materials and methods

### Source of data

Data from the Ethiopian Demographic and Health Surveys 2016 were used for this analysis. The EDHS 2016 survey was organized to allow estimates of key indicators for the country as a whole, for urban and rural areas separately, and for each of the nine regions and the two administrative cities. Each region was stratified into urban and rural areas, yielding 21 sample strata. Samples of Enumeration Areas (EAs) were selected independently in two stages in each stratum. In the first stage, based on the 2007 PHC, an independent selection was implemented in each sampling stratum involving a total of 645 EAs (202 in urban and 443in rural) areas with probability proportional to EA size. In the second stage, a fixed number of 28 households per cluster were selected through an equal probability systematic selection from the newly created household listings.

### Outcome variable

Unemployment status of women was the dependent variable containing two categories: unemployed women and employed women. According to ILO's definition, people who are simultaneously "without work", "currently available for work" and "seeking work" are considered as unemployed. Thus, the outcome for the $i^{th}$ woman is represented by a random variable $Y_i$ with two possible values coded as 1 and 0. In view of this, the outcome of the $i^{th}$ woman $Y_i$ was measured as a dichotomous variable.

$$Y_i = \begin{cases} 1, & if the i^{th} womem is unemployed \\ 0, & otherwise \end{cases}$$

### Data management and statistical analysis

Data were extracted and re-categorized using SPSS version 21 software.Spatial analyses were performed using Geoda V.1.8.10 (geode center.github.ib), QGIS V.2.18.0 (qgis.org) and Arch GIS software V.10.1 (arcgis.com), and base files of the administrative regions for Ethiopia were obtained from DIVA (diva-gis.org). Multilevel logistic regression model was made using R software version 3.5.3.

  **Spatial analysis.** Spatial analysis was conducted by joining the occurrence of unemployment (as proportions) with each cluster to the corresponding geospatial location (survey cluster values). The values of Demographic Health Survey data were merged with the geographic positioning system (GPS) dataset in Geoda software, and the values were imported into the QGIS software. Proportions of unemployment were then computed at lower (cluster), zonal,

and regional levels using QGIS." [23]. The spatial patterns of the rate of unemployment among women were visualized, and a spatially smooth proportion was obtained through empirical Bayes estimation method. The smoothed proportions presented clearer patterns in contexts where the problem was most severe. The spatial empirical Bayes 'smooth' estimates technique was applied for the spatial heterogeneity. The estimation technique guarantees that estimates in the neighboring states are more alike than estimates in states that are further away [24]. Getis-OrdGi* statistics was used for this spatial analysis. Local Getis-OrdGi* statistics helped identify the hot and cold spotted areas for unemployment in women using GPS latitude and longitude coordinate readings that were taken at the nearest community center for EAs or EDHS 2016 clusters [25].Using Kuldorff'sSaTScan version 9.6 software, we used a Bernoulli model spatial Scan statistics to know the locations of statistically significant clusters for unemployment rate [23]. This model is used when unemployment are in the study area, as is the case for employed as controls, because of the scanning window which moves across the area. It was possible to detect small and large clusters which contained more than themaximum limit with the circular shape of the window, using the default maximum spatial cluster size of less than half the population. P- Values and log-likelihood ratios

**Multilevel logistic regression analysis.** The 2016 EDHS data is hierarchical in its structure andwomen are nested within enumeration area (communities). Hence, considering the hierarchical nature of the data, multilevel logistic regression models were applied to identify factors associated with women unemployment [26].The log of the probability of women unemployment was modeled using a two-level multilevel model as follows:

$$Log\left[\frac{\pi_{ij}}{1-\pi_{ij}}\right] = \beta_0 + \beta_1 X_{ij} + \beta_2 Z_{ij} + u_j + e_{ij}$$

Where, $i$ and $j$ are the level 1 (individual) and level 2 (community) units, respectively; $X$ and $Z$ refer to individual and community-level variables, respectively; $\pi_{ij}$ is the probability of unemployment for the $i^{th}$ women in the $j^{th}$ community; and the $\beta$ indicates the fixed coefficients. $\beta_0$ is the intercept-the effect on the probability of women unemployment in the absence of influence of predictors; $u_j$ showed the random effect (effect of the community on women unemployment for the $j^{th}$ community, and $e_{ij}$ showed random errors at the individual level. Assuming each community had different intercepts ($\beta_0$) and fixed coefficient ($\beta$), the hierarchical (clustered) nature of the data and the within and between community variations were taken into account. Four models were fitted to identify community and individual level factors associated with women unemployment. The first model (Model 1 or empty model) contained no explanatory variables. Instead, it was fitted to decompose the total variance into its individual- and community-level components. Individual-level factors were incorporated in the second model. In the third model, house hold level factors were included. In the fourth model, community-level factors were included. Finally, individual and community-level factors were included in the fourth model. Model comparison was made using deviance information criteria (DIC), Akaike's Information Criterion (AIC), and Bayesian's Information Criterion (BIC). The model with the smallest value of the information criterion was selected as the final model of the analysis [27]. For the result of fixed effect, odds ratio (ORs) with 95% confidence intervals (CIs) determined the statistical significance. The P-value of $\leq 0.05$ is considered statistically significant.The measures of variation (random-effects) were summarized using ICC, Median Odds Ratio (MOR) and proportional change in variance (PCV) to measure the variation between enumeration areas (clusters). ICC, which is a measure of within-cluster and variation between individuals within the same cluster was calculated using the formula:
$ICC = \frac{V_A}{V_A + \pi^2/3} = \frac{V_A}{V_A + 3.29}$, where $V_A$ is the estimated variance in each model described elsewhere

[28]. The total variation attributed to individual or/and community level factors at each model was measured with a proportional change in variance (PCV)calculated as: PCV $= \frac{V_A - V_B}{V_A}$, where $V_A$ = variance of the initial model, and $V_B$ = variance of the model with more terms [28]. The MOR is the median odds ratio between a person with a higher propensity and a person with a lower propensity that compares two people from two different randomly chosen clusters and measures unexplained cluster heterogeneity as well as variation between clusters by comparing two people from two different randomly chosen clusters. It was calculated with the following formula:

$$MOR = \exp(\sqrt{2 * V_A * 0.6745}) \approx \exp(0.95\sqrt{V_A}),$$ where $V_A$ is the cluster level variance [28,29]. The MOR measure is always greater than or equal to 1. If the *MOR* is 1, there is no variation between clusters [30].

## Ethical consideration

Publicly available EDHS 2016 data were used for this study. Informed consent was taken from each participant, and all identifiers were removed.

## Results

### Socio-demographic characteristics of study participants

A total of 15,683 women were included in the study. An unemployment rate of about 29.8% was observed among uneducated women. 48.9% of unemployment occurred in women with secondary education and above. Less than half (45.8%) of the unemployment were attributed to rich women. The unemployment of urban and rural women was 50.7% and 28.7%, respectively. Married women (32.5%) had the lowest unemployment. While the unemployment rate of uneducated husbands was 24.4%, husbands with educational level of secondary and above had an unemployment rate of 42.7%. The rich women had an unemployment rate of 45.8% (**Table 1**).

### Incremental spatial autocorrelation

A peak in the graph denotes the distance at which the clustering is most pronounced. The color of each point on the graph corresponds to the statistical significance of the z-score values. The incremental spatial autocorrelation demonstrated that with 10 distance bands beginning at 2 km, women unemployment clustering was detected at 2.94 km distance. Statistically, a significant z-score (10.55) indicates that spatial clustering ofwomen unemployment was most pronounced at 2.94 Km distance (Fig 1).

### Spatial pattern of women unemployment

In Ethiopia, the spatial distribution of women's unemploymentwas spatially clustered with Global Moran's I = 0.064298 (z-score = 16.375386, P-value <0.0001). This demonstrated that spatial hotspot and coldspot clustering was identified in Ethiopian regions. Given the z-score of 16.375386, there was less than a 1% chance that this high-clustered pattern was the result of randomchance. The bright red and blue colors on the tails indicate a higher level of significance (Fig 2).

### Hot spot (Getis-Ord Gi*) analysis

Fig 3 summarizes a hot and cold spot analysis of the risk locations for women's unemployment in Ethiopia. Beneshagul and Gambela, Oromia, WesternAmhara, and northeren SNNPR are

**Table 1. Socio-demographic characteristics of study participants.**

| | | Women Unemployment rate | |
|---|---|---|---|
| **Variables** | **Categories** | **No (%)** | **Yes (%)** |
| Women education level | No education | 4937(70.2) | 2096(29.8) |
| | Primary | 3317(63.6) | 1896(36.4) |
| | Secondary and above | 1757(51.1) | 1680(48.9) |
| Husband education level | No education | 3352(75.6) | 1079(24.4) |
| | Primary | 1961(64.2) | 1093(35.8) |
| | Secondary and above | 4698(57.3) | 3500(42.7) |
| Marital status | Living alone | 2722(63.6) | 1556(36.4) |
| | Married | 6636(67.5) | 3188(32.5) |
| | Other | 653(41.3) | 928(58.7) |
| Economic status | Poorest | 4398(74) | 1542(26) |
| | Middle | 1416(70.7) | 586(29.3) |
| | Richest | 4197(54.2) | 3544(45.8) |
| Residence | Rural | 7374(71.3) | 2961(28.7) |
| | Urban | 2637(49.31) | 2711(50.7) |
| Age of women | <25 years | 4500(70.3) | 1901(29.7) |
| | 25–34 years | 3020(59.4) | 2066(40.6) |
| | >34years | 2491(59.4) | 1705(40.6) |
| Household size | 0–5 | 5045(59.2) | 3474(40.8) |
| | 6–10 | 4629(68.9) | 2086(31.1) |
| | >10 | 337(75.1) | 112(24.9) |
| Sex of household | Female | 2600(53.8) | 2230(46.2) |
| | Male | 7411(68.3) | 3442(31.7) |

the regions with the highest unemployment rates. Tigray, Afar, Dire Dawa, Hariri, northern Amhara, and Somalia on the other hand, have been designated as cold-spot (low risk) regions (Fig 3).

## Spatial SaTScan analysis of unemployment women across region

Most likely (primary clusters) and secondary clusters of unemployment women were identified. In 2016 EDHS, spatial scan statistics identified a total of high and modest performing spatial clusters of unemployment women. A total of 164 significant clusters were identified. Of these, among the significant clusters, 120 were most likely (primary cluster) and the other 44 were secondary.The spatial window of the primary cluster was located at Tigray, Amhara, southern Gambela,eastern Oromia and Afar which was centered at (12.376936 N,38.357984 E) / 318.07 km, RR = 1.43, and log likelihood ratio (LLR) of 200.78 at p-value < 0.0001. It showed that house holds in the spatial window had 1.43 times higher unemployment women ratethan those outside the window (S1 Table and Fig 4).

## Spatial interpolation

The spatial kriging interpolation analysis predicted high risk regions for women Unemployment in Ethiopia. Predication of high risk areas were indicated by Blue color. Beneshagul, western and central Oromia regional state were predicted as more risky areas compared to other regions. women in this areas were endangered to unemployment in Ethiopia. In other hand; women in Somalia, Afar, Eastern Amhara, Dire Dawa, central Tigray and eastern Oromia were identified as vulnerable to poor unemployment in Ethiopia (Fig 5).

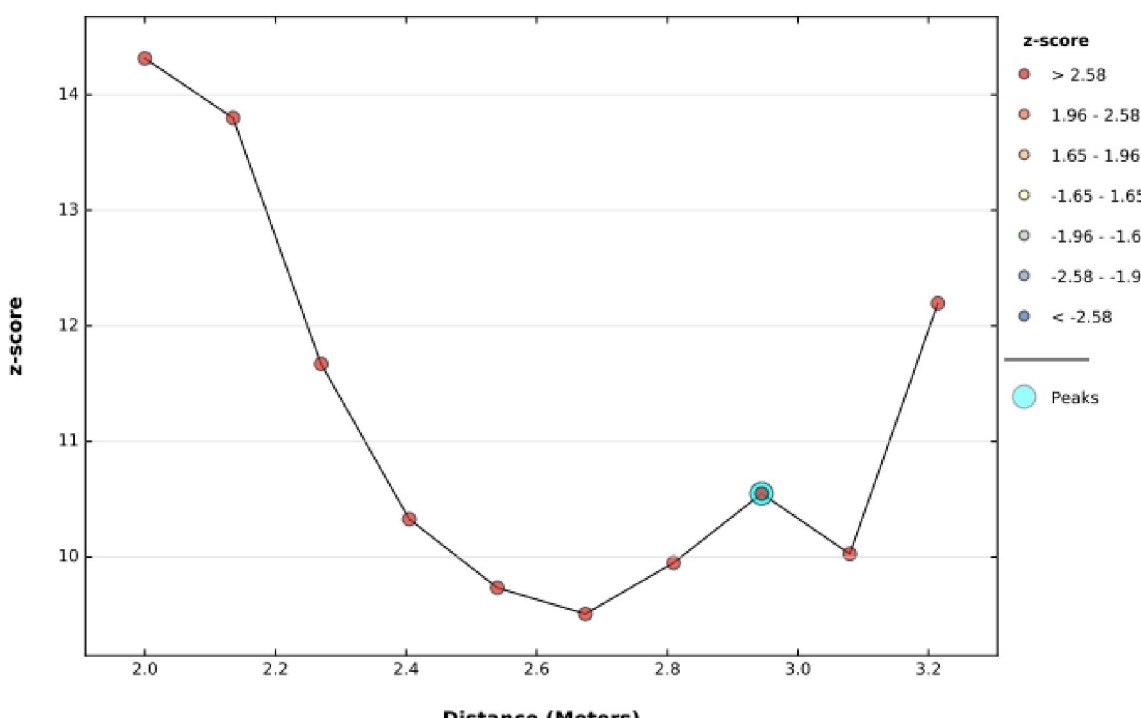

**Fig 1. Incremental spatial autocorrelationof women unemployment status in Ethiopia, 2016 EDHS.**

## Factors associated with women unemployment in Ethiopia

Multivariable-multilevel logistic analysis revealed that residence, age of women, marital status, women education level, husband/partners level of education, husband/partners occupation, sex of the household's head, wealth index, family size, region and pregnancy were statistically significant factors for women unemployment. Women with a primary level of education were 0.88 (AOR = 0.88, 95%CI: 0.80, 0.98) times less likely to be unemployed than women who were not educated. Women whose level of education was secondary and above were 0.71 (AOR = 0.71, 95%CI: 0.62, 0.82) times less likely to be unemployed than women who have no education. Women whose husbands had a primary education level were 0.82 (AOR = 0.82, 95%CI: 0.73, 0.92) times less likely to be unemployed than women whose husbands were not educated. Rich women were 0.79 (AOR = 0.79, 95% CI: 0.70, 0.90) times less likely to be unemployed than poor women. Compared with women aged less than 25 years, the likelihood of unemployment among women aged 25–34 years was 0.50 (AOR = 0.50, 95% CI: 0.44, 0.54) times lower. The likelihood of unemployment for a family whose size was 6–10 was 1.2 (AOR = 1.2, 95% CI: 1.11,1.31) times higher than those whose family size was less than 6. Women who are already pregnant were1.24 (AOR = 1.24, 95% CI: 1.06, 1.5) times more likely to be unemployed than women who were not pregnant. Women living in households headed by men were 1.40 (AOR = 1.4, 95% CI: 1.28,1.50) times more likely to be unemployed than women living in households headed by women. Women with non-agricultural husbands were 0.47(AOR = 0.47, 95% CI: 0.30, 0.70) times less likely to be unemployed than women whose husbands are unemployed. The odds of unemployment for women living in Afar (AOR = 2.2,

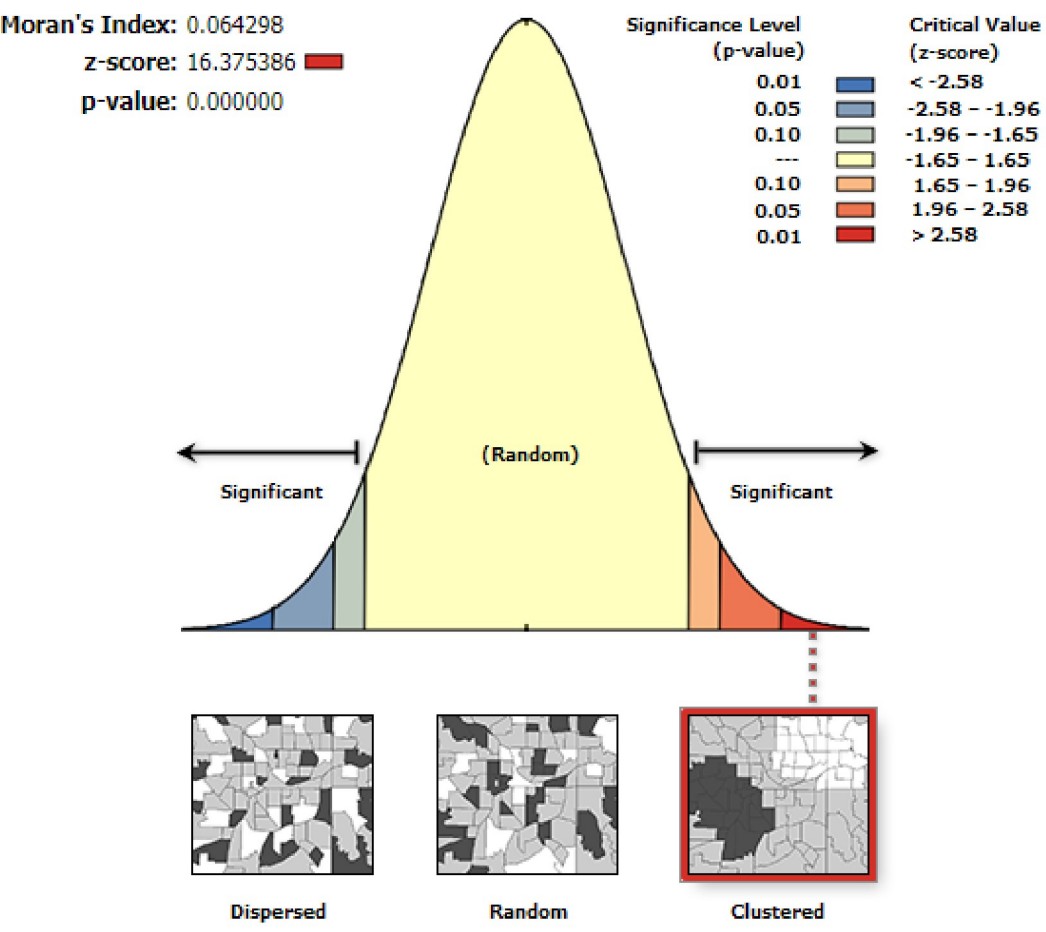

Given the z-score of 16.3753858906, there is a less than 1% likelihood that this clustered pattern could be the result of random chance.

**Fig 2. Spatial autocorrelation analysis of women unemployment status in Ethiopia, 2016 EDHS.**

95% CI: 1.62, 2.99), Amhara (AOR = 1.70, 95% CI: 1.30, 2.20), Somali(AOR = 1.90, 95% CI: 1.45, 2.60), and Dire Dawa(AOR = 1.44, 95% CI: 1.06, 1.96) was higher than women living in Tigray. Women living in Benishangule-Gumuz were 0.43 (AOR = 0.43, 95% CI: 0.32, 0.60) times less likely to be unemployed than women living in Tigray. Women living in urban areas were 0.60 (AOR = 0.60, 95% CI: 0.50, 0.70) times less likely to be unemployed than women living in rural areas (Table 2)

## Random effect analysis (Measures of variation)

The findings of the random logistic regression analysis are summarized in Table 3. The empty model (Model I) shows that there are discrepancies in the community's unemployment rates for women. Women's unemployment rates varied by roughly 21.48 percent due to community-level factors (ICC = 22.48 percent). In the null model, women's unemployment had the highest MOR value (2.46), showing that there was variation between communities clustered as MOR was 2.46 times more than the comparison (MOR = 1). Furthermore, the PCV of the whole model (model IV) revealed that individual and community factors accounted for roughly 59 percent of the difference in women's unemployment among populations. The

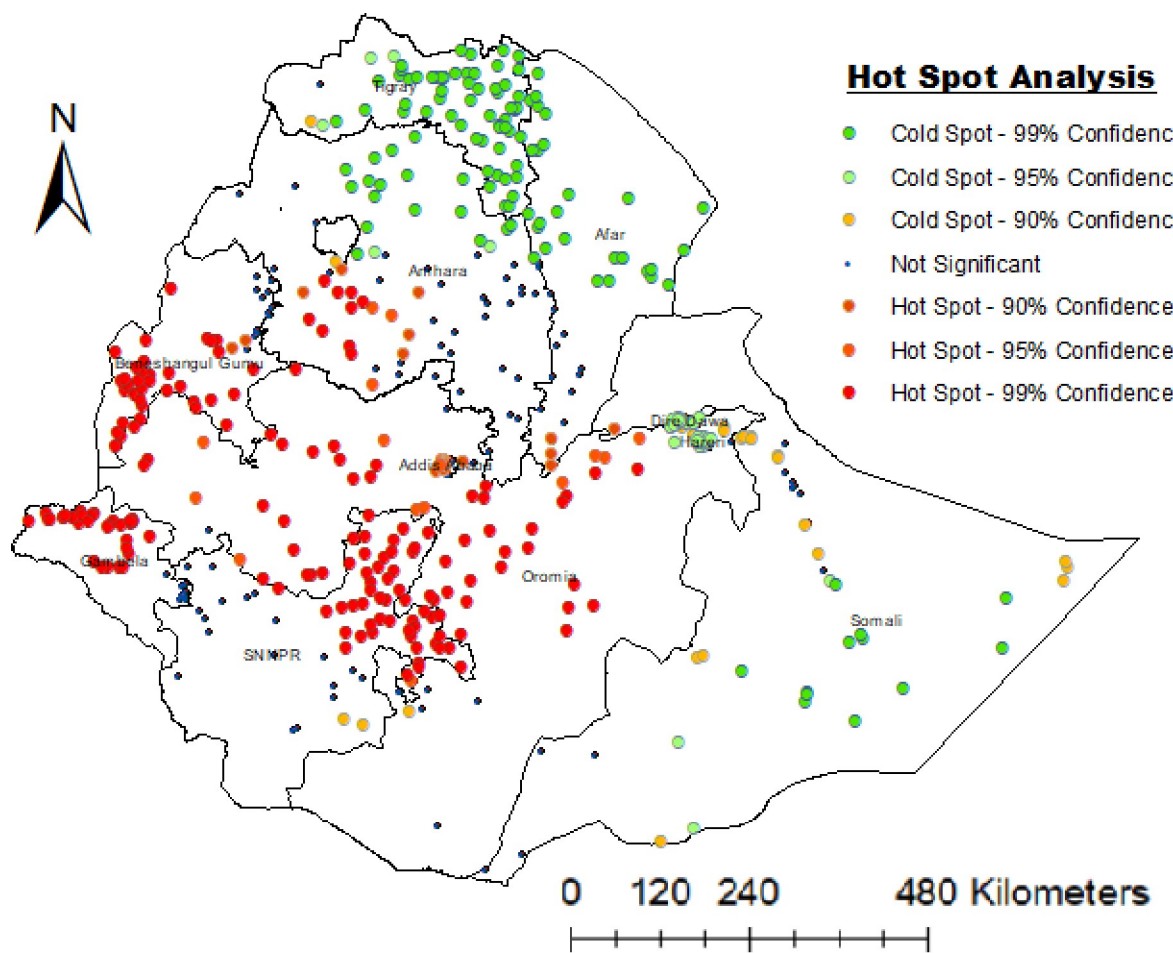

**Fig 3. Hot spot and cold spot identification of unemployment women in Ethiopia, 2016 EDHS (Source of shapefiles: https:// africaopendata.org/dataset/ethiopia-shapefiles).**

unexplained community variance in women unemployment was decreased to MOR of 1.78 when all factors were incorporated to the null model. When all factors are taken into account, the effect of clustering remains statistically significant in the overall model (Table 3).

## Discussion

In Ethiopia, approximately 63.8 percent of women are unemployed, with rates varying by location. The distribution of women's jobless status is clustered, showing that it is not random in Ethiopia, according to the spatial analysis. The western, north-west, central and south-western regions of Ethiopia are identified as high-risk (hot spot) areas for women's unemployment rates, whereas the northern, south eastern, and north easternares are identified as low-risk areas. In the regions of Tigray, Amhara, southern Gambela,eastern Oromia and Afar, SaTscan identified significant primary clusters (most likely clusters).This could be because the bulk of people living in the this area are pastoralists. A similar result was found in Ethiopia [19] and Turkey [20]. Ethiopia's unemployment rate depends on geographic location and gender, as well as public policies and directions to combat unemployment and its' harmful consequences. Although Ethiopia's economy has shown impressive declines in unemployment, women have not benefited as much. In fact, women have higher unemployment rates [31–33].

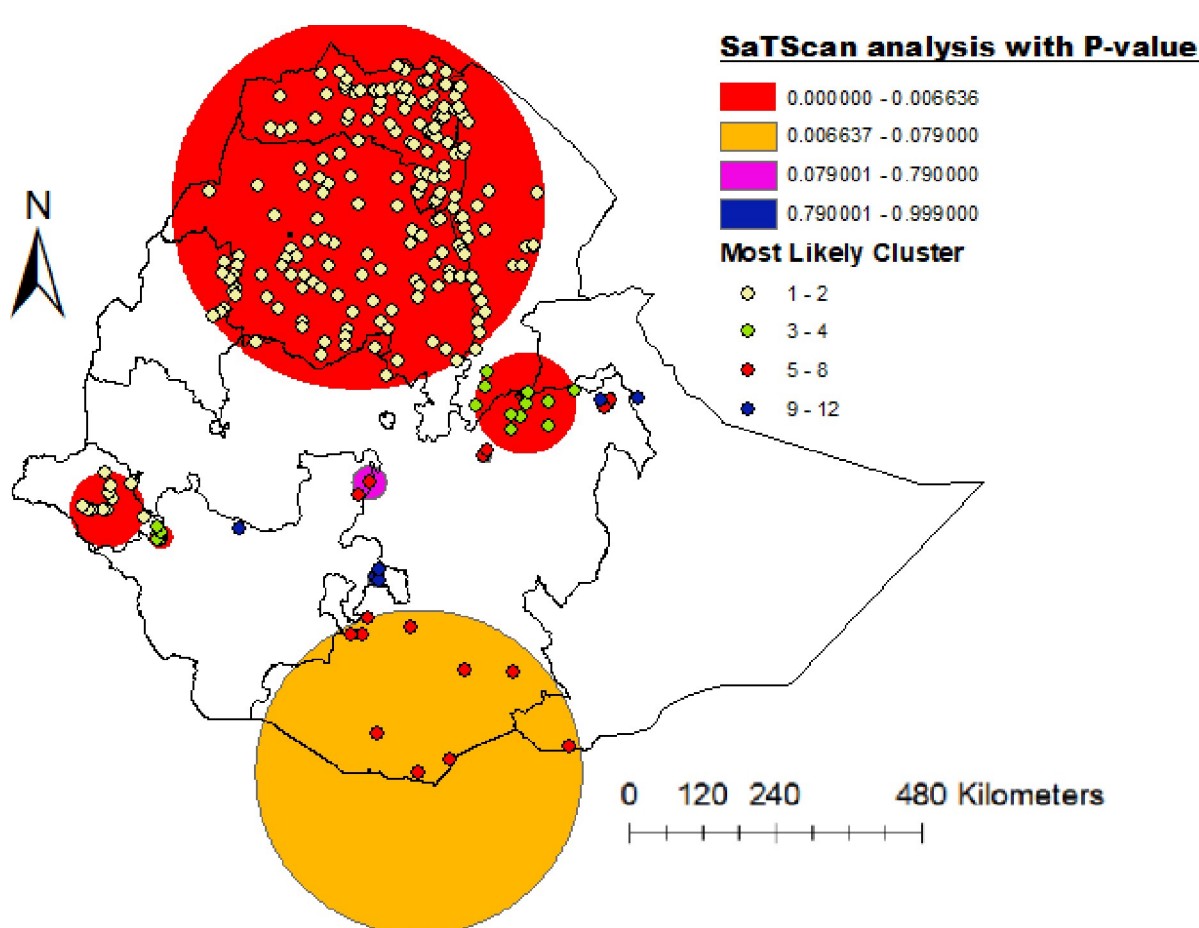

**Fig 4. SaTScanAnalysisof unemployment women in Ethiopia, 2016 EDHS (Source of shapefiles: https://africaopendata.org/dataset/ethiopia-shapefiles).**

Individual and community-level factors accounted for around 58.89 percent of the variance in women's unemployment, according to the results of the random effects model. In contrast to illiterate women, educated women were less likely to be jobless. Women with educated spouses had a higher rate of unemployment than women with uneducated husbands. Previous findings [7,19,31,32] are supported by this. People with a higher level of education have a better chance of landing a good job. Furthermore, it empowers women to make their own decisions, to be accepted by their family and society, and to have more job prospects. Unemployment was lower among pregnant women than among non-pregnant women. This finding is consistent with earlier research [1,19,33], which found that pregnant women were less likely than non-pregnant women to work in a given year.

Women who lived alone were more likely than widowed or divorced women to be unemployed. A study in Ethiopia's Harari Region [1], Ethiopia [19,32], and Pakistan's Sahiwal District [2] supports this conclusion. Female-headed families had a lower unemployment rate than male-headed households. A research report from Halaba town, in Southern Ethiopia (SNNPR) [32] supports this conclusion. Women's increased participation in household development activities may be to blame for these phenomena.

Women from wealthy families were less likely than those from poor families to be unemployed. This finding is in line with earlier research from Ethiopia [18,19], South Africa [34],

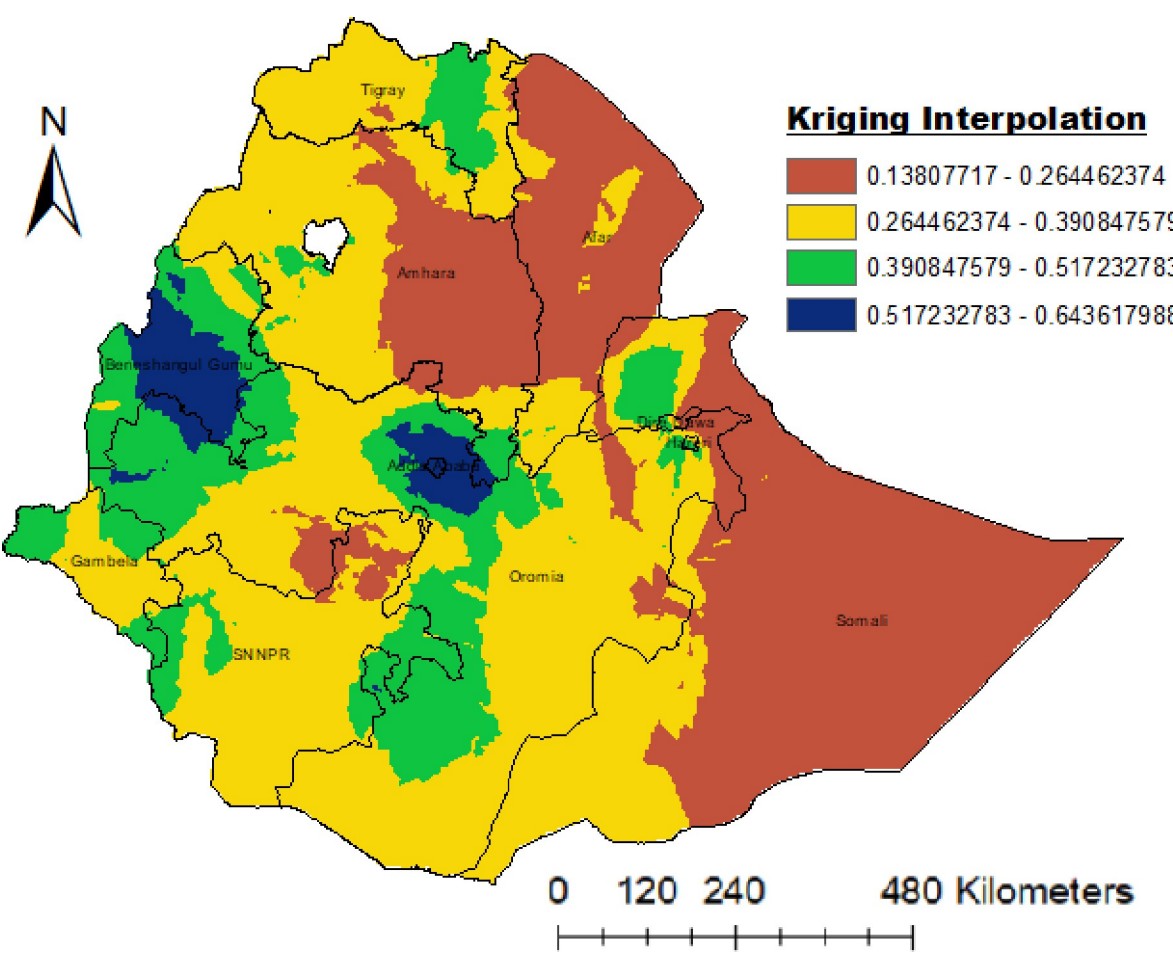

**Fig 5. Kriging interpolation of unemployment women in Ethiopia,2016 EDHS (Source of shapefiles: https://africaopendata.org/dataset/ethiopia-shapefiles).**

and Ghana [35], all of which found that women from low-income households had the greatest unemployment rate. This could be because women from higher-income families may have better job-searching resources or access to the first capital needed to start their own business.

Women in urban were less likely than those in rural areas to be unemployed. This conclusion is supported by research from Ethiopia [19] and South Africa [36]. This could be due to the high shadow value of home production activities among women in rural areas. Unemployment was higher among women under the age of 25 than among older women. This finding is consistent with a study done in Urban Districts, Harari Region, Ethiopia [1], Halaba town, SNNPR, Southern Ethiopia [34], Ethiopia [19] and Sahiwal District, Pakistan [2] which found that women in the youngest age group studied are most influenced by unemployment.

Women's unemployment rates were greater in households with bigger family sizes than in homes with smaller families. This finding is consistent with findings from studies conducted in Harari Region, Ethiopia [1], Halaba Town, SNNPR, Southern Ethiopia [32], Bahir Dar City, Northwest, Ethiopia [18], and Sahiwal District, Pakistan [2], which found that women with families of five or more were more likely to be unemployed than women with families of less than five. The unemployment rate of women whose husband has a non-agricultural employee was less than women whose husband was unemployed. Women whose husband has an

**Table 2. Factors associated with women unemployment in Ethiopia, EDHS 2016.**

| Variable | Null | Model 1 | Model 2 | Model 3 |
|---|---|---|---|---|
| | OR(95%CI) | OR(95%CI) | OR(95%CI) | OR(95%CI) |
| **Marital status** | | | | |
| Living alone | | 1 | | 1 |
| Married | | 1.09(0.96,1.25) | | 1.08(0.94,1.24) |
| Other | | 0.53(0.45,0.62)* | | 0.51(0.43,0.613)* |
| **Mother education level** | | | | |
| No Education | | 1 | | 1 |
| Primary | | 0.823(0.74,0.91)* | | 0.88(0.80,0.98)* |
| Secondary and above | | 0.65(0.57,0.74) * | | 0.71(0.62,0.82) * |
| **Husband education level** | | | | |
| No Education | | 1 | | 1 |
| Primary | | 0.78(0.691,0.88)* | | 0.82(0.73,0.92)* |
| Secondary and above | | 0.83(0.72,0.96)* | | 0.89(0.77,1.03) |
| **Wealth index** | | | | |
| Poor | | 1 | | 1 |
| Middle | | 0.88(0.77,1.01) | | 0.95(0.83,1.09) |
| Rich | | 0.64(0.57,0.72)* | | 0.79(0.70,0.91)* |
| **Drug addiction** | | | | |
| No | | 1 | | 1 |
| Yes | | 0.96(0.87,1.05) | | 0.97(0.88,1.06) |
| **Age of women in year** | | | | |
| 15–24 | | 1 | | 1 |
| 25–34 | | 0.459(0.414,0.5)* | | 0.5(0.44,0.54)* |
| 34–49 | | 0.457(0.41,0.51)* | | 0.49(0.44,0.62)* |
| **Family size** | | | | |
| 2–5 | | 1 | | 1 |
| 6–10 | | 1.24(1.14,1.34)* | | 1.20(1.11,1.31)* |
| >10 | | 1.57(1.21,2.02)* | | 1.51(1.13,1.9)** |
| **Child under age of 5 years** | | | | |
| No | | 1 | | 1 |
| Yes | | 1.51(0.82,8.20) | | 1.62(0.34,88) |
| **Sex of household head** | | | | |
| Female | | 1 | | 1 |
| Male | | 1.39(1.27,1.27)* | | 1.40(1.28,1.51)* |
| **Pregnancy** | | | | |
| No | | 1 | | 1 |
| Yes | | 1.27(1.08,1.48)** | | 1.24(1.06,1.50)* |
| **Migration status** | | | | |
| Visitors | | 1 | | 1 |
| Usual Residence | | 0.95(0.75,1.22) | | 0.97(0.76,1.23) |
| **Husbands occupation** | | | | |
| Not Working | | 1 | | 1 |
| Agric Employee | | 0.45(0.3,0.7)* | | 0.47(0.31,0.70) * |
| Non-Agric Employee | | 0.6(0.5,0.72) * | | 0.63(0.52,0.71) * |
| **Region** | | | | |
| Tigray | | | 1 | 1 |
| Afar | | | 2.70(2.00,3.71) * | 2.21(1.62,2.99) * |

(*Continued*)

**Table 2.** (Continued)

| Variable | Null | Model 1 | Model 2 | Model 3 |
|---|---|---|---|---|
| Amhara | | | 1.67(1.26,2.23) * | 1.70(1.31,2.22) * |
| Oromia | | | 1.06(0.81,1.38) | 0.95(0.74,1.23) |
| Somali | | | 2.52(1.9,3.36) * | 1.92(1.45,2.61) * |
| Benishangul-Gumuz | | | 0.49(0.37,0.66)* | 0.43(0.32,0.6) * |
| SNNP | | | 1.02(0.78,1.33) | 0.921(0.71,1.19) |
| Gambela | | | 0.964(0.72,1.3) | 0.91(0.72,1.21) |
| Harari | | | 1.18(0.86,1.61) | 1.10(0.81,1.53) |
| Dire Dawa | | | 1.56 (1.14,2.14)* | 1.44(1.06,1.96) * |
| Addis Ababa | | | 0.81(0.6,1.094) | 0.80(0.62,1.04) |
| **Residence** | | | | |
| Rural | | | 1 | 1 |
| Urban | | | 0.37(0.32,0.44)* | 0.63(0.52,0.71) * |
| **Exposed to Mass media** | | | | |
| No | | | 1 | 1 |
| Yes | | | 1.14(0.99,1.30) | 1.12(0.95,1.27) |

1 reference category for categorical variable and * reference P-value < 0.0001.

agricultural employment were less likely to be unemployed than women whose husband was unemployed. This is supported by findings from other studies in Ethiopia [1] and Spain [35].

Furthermore, women's unemployment was significantly influenced by their geographic location. Women in the Afar, Amhara, Somalia, and Dire Dawa areas had greater unemployment rates than women in the Tigray region. The findings of this study are consistent with those of earlier Ethiopian studies [19]. Women who have access to the media have a lower chance of becoming unemployed than women who do not. This finding is backed up by findings from a research report conducted in Halaba, Southern Ethiopia [32].

## Conclusion

The study revealed that about 63.8% of Ethiopian women are unemployed and the unemployment rate varied across the regions. The spatial analysis indicated that the distribution of women's unemployment status is clustered indicating that it was not random. High-risk (hot spot) areas for women's unemployment rates were in the eastern, north-eastern and south-eastern

**Table 3. Measures of variation and model fit statistics on women unemployment in Ethiopia.**

| Measures of variations | Model 1 | Model 2 | Model 3 | Model 4 |
|---|---|---|---|---|
| Variance | 0.90(0.77, 1.05)* | 0.63(0.51, 0.70)* | 0.42(0.35,0.50)* | 0.37(0.30,0.45)* |
| ICC (%) | 21.48 | 16.07 | 11.32 | 10.11 |
| PCV (%) | Reference | 30.00 | 53.33 | 58.89 |
| MOR | 2.46 | 2.13 | 1.85 | 1.78 |
| **Model fit statistics** | | | | |
| DIC (-2log likelihood) | 19127.6 | 18245.9 | 18778.122 | **18033.86** |
| AIC | 19131.59 | 18287.9 | 18806.12 | **18099.86** |
| BIC | 19146.91 | 18448.77 | 18913.37 | **18352.65** |

*reference P-value < 0.0001.

parts of Ethiopia, while low-risk areas were in the northern, southern and western Ethiopia. Residence, age, marital status, educational level, husband's education level, husband occupation, sex of household head, wealth index, family size, region and pregnancy were statistically important factors affecting women's unemployment. The unemployment rate of women in Ethiopia differed from cluster to cluster. Hence, improving entrepreneurship and women's education, sharing business experiences, supporting entrepreneurs could be useful measures to reduce women's unemployment. Furthermore, community-based programs that prioritize the participation of poor women and improve their access to the media and the labor market need to be developed.

## Supporting information

**S1 Table. Significant SaTScan spatial scan clusters for unemployment women across region in Ethiopia, 2016 EDHS.**
(DOCX)

## Acknowledgments

We would like to thank the Ministry of Health and Central Statistical Agency, Government of Ethiopia, for making the data freely available for research purposes.

## Declaration

### Ethics approval and consent to participate

This study is a secondary data analysis of the EDHS, which is publicly available. Approval to use the data was sought from MEASURE DHS/ICF International, and permission was granted for its use. The original DHS data were collected in conformity with international and national ethical guidelines. Ethical clearance was provided by the Ethiopian Public Health Institute (EPHI) (formerly the Ethiopian Health and Nutrition Research Institute (EHNRI) Review Board, the National Research Ethics Review Committee (NRERC) at the Ministry of Science and Technology, the Institutional Review Board of ICF International, and the United States Centers for Disease Control and Prevention (CDC). Written consent was obtained from mothers/caregivers and data were recorded anonymously at the time of data collection during the EDHS 2016

## Author Contributions

**Conceptualization:** Solomon Sisay Mulugeta, Shewayiref Geremew Gebremichael.

**Data curation:** Solomon Sisay Mulugeta.

**Formal analysis:** Solomon Sisay Mulugeta.

**Investigation:** Solomon Sisay Mulugeta, Setegn Muche Fenta.

**Methodology:** Solomon Sisay Mulugeta, Setegn Muche Fenta.

**Resources:** Solomon Sisay Mulugeta.

**Software:** Solomon Sisay Mulugeta, Setegn Muche Fenta.

**Supervision:** Solomon Sisay Mulugeta, Shewayiref Geremew Gebremichael.

**Validation:** Solomon Sisay Mulugeta, Shewayiref Geremew Gebremichael, Setegn Muche Fenta.

**Visualization:** Solomon Sisay Mulugeta, Setegn Muche Fenta, Berhanu Engidaw Getahun.

**Writing – original draft:** Solomon Sisay Mulugeta, Setegn Muche Fenta.

**Writing – review & editing:** Solomon Sisay Mulugeta, Shewayiref Geremew Gebremichael, Setegn Muche Fenta, Berhanu Engidaw Getahun.

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
