## [Decision Letter · Decision Letter 0]

28 Jul 2021

PONE-D-21-08056

Geographical variation and determinants of women unemployment status in Ethiopia; A multilevel and spatial analysis from 2016 Ethiopia Demographic and Health Survey data

PLOS ONE

Dear Dr. Fenta,

Thank you for submitting your manuscript to PLOS ONE. After careful consideration, we feel that it has merit but does not fully meet PLOS ONE’s publication criteria as it currently stands. Therefore, we invite you to submit a revised version of the manuscript that addresses the points raised during the review process.

We look forward to receiving your revised manuscript.

Kind regards,

Ceyhun Elgin

Academic Editor

PLOS ONE

Additional Editor Comments (if provided):

You can read the reviewer letters below. As you can see one reviewer is positive and suggests some minor improvements, whereas another reviewer is more skeptical of what you are doing and recommended a rejection. I am giving you a chance to revise your paper (it is a major revision though), however, for this paper to be accepted; you have to convince the reviewer 2.

Journal Requirements: 

2. Please amend either the abstract on the online submission form (via Edit Submission) or the abstract in the manuscript so that they are identical.

4. We note that Figures 4.1,4.2 and 4.3 in your submission contain map images which may be copyrighted. All PLOS content is published under the Creative Commons Attribution License (CC BY 4.0), which means that the manuscript, images, and Supporting Information files will be freely available online, and any third party is permitted to access, download, copy, distribute, and use these materials in any way, even commercially, with proper attribution. For these reasons, we cannot publish previously copyrighted maps or satellite images created using proprietary data, such as Google software (Google Maps, Street View, and Earth). For more information, see our copyright guidelines: http://journals.plos.org/plosone/s/licenses-and-copyright.

A. You may seek permission from the original copyright holder of Figures 4.1,4.2 and 4.3 to publish the content specifically under the CC BY 4.0 license.  

B. If you are unable to obtain permission from the original copyright holder to publish these figures under the CC BY 4.0 license or if the copyright holder’s requirements are incompatible with the CC BY 4.0 license, please either i) remove the figure or ii) supply a replacement figure that complies with the CC BY 4.0 license. Please check copyright information on all replacement figures and update the figure caption with source information. If applicable, please specify in the figure caption text when a figure is similar but not identical to the original image and is therefore for illustrative purposes only.

5. Thank you for submitting the above manuscript to PLOS ONE. During our internal evaluation of the manuscript, we found significant text overlap between your submission and the following previously published works.

- https://bmcpublichealth.biomedcentral.com/articles/10.1186/s12889-019-7529-z

- https://bmjopen.bmj.com/content/9/4/e027276

- https://pubmed.ncbi.nlm.nih.gov/24411023/

Please revise the manuscript to rephrase the duplicated text, cite your sources, and provide details as to how the current manuscript advances on previous work. Please note that further consideration is dependent on the submission of a manuscript that addresses these concerns about the overlap in text with published work.

Reviewers' comments:

Reviewer's Responses to Questions

**Comments to the Author**

1. Is the manuscript technically sound, and do the data support the conclusions?

Reviewer #1: Partly

Reviewer #2: Partly

2. Has the statistical analysis been performed appropriately and rigorously? 

Reviewer #1: Yes

Reviewer #2: No

3. Have the authors made all data underlying the findings in their manuscript fully available?

Reviewer #1: Yes

Reviewer #2: Yes

4. Is the manuscript presented in an intelligible fashion and written in standard English?

Reviewer #1: No

Reviewer #2: No

5. Review Comments to the Author

Reviewer #1: I am glad to review and assess this interesting article, entitled, Geographical variation and determinants of women unemployment status in Ethiopia; A multilevel and spatial analysis from 2016 Ethiopia Demographic and Health Survey data. The organization of this article is good and satisfactory. The Introduction part and methodology portions are adequate. I suggest the authors improve the Introduction section by adding some latest articles' citations to enhance the work quality.

Overall, the manuscript is a good piece of work. I recommend that authors do a little more work and add the latest literature to support the study, as suggested. The English level is not good and smooth, e.g., the language standard, specifically the grammar, of not sufficient quality to meet scientific merit for publication. I accept this manuscript after minor revision, as I have recommended.

Reviewer #2: This work contains two simple empirical exercises about female unemployment in Ethiopia.

In the first one, the authors routinely use standard spatial techniques to detect clusters of areas with low or high unemployment. While technically well carried out (mainly because it is simply implemented in usual software), results are barely commented and it is particularly unclear the contribution of this analysis. The potential contribution should be focused on commenting about spatial correlation. To start with, both Figure 4.2 and 4.3 are barely commented. The terms used in Figure 4.2 are confusing: "High Cluster" can also apply to what the authors call "High outlier", because (1) both are clusters (but in the latter case this is not mentioned), (2) the term outlier is not appropriate because being a high-low area is NOT weird (=outlier), but a simple indication of negative spatial correlation. If the authors could just READ their software output (labels in the picture their software provided), they could find the standard term: high-high cluster and high-low cluster. In any case, authors just provide a purely descriptive minimum comments on the areas with high or low unemployment incidence, but no discussion at all about the implications or how these results can be useful. Commenting that female unemployment is different in different clusters is very vague and can be concluded from any other study. And the general recommendations in the last two sentences of the Conclusions are NOT directly supported by the analysis.

Therefore, it is unclear the value added by the spatial analysis. The authors do not seem to feel that it was useful, because the second part of their work does not use the information about spatial correlation among clusters (enumeration areas). They just apply a multilevel logistic model, but they do not seem to take into account spatial correlation to compute standard errors (and therefore confidence interval). It is, of course, important to account for both individual level and community/clusters level factors to avoid biased results (i.e., as the authors mention, incorrectly concluding that a factor is significant or not). But the spatial correlation analysis is not needed to account for this. If one finds that spatial correlation exists, this should be incorporated into the subsequent analysis. In their current analysis, the authors allow for within-cluster (community) correlation, but assume no between-clusters correlation despite of their previous findings!

The (logistic) regression analysis is just a sophisticated but ultimately simple exercise about which variables are correlated with observed female unemployment. On the one hand, no really new result is found. The authors again present the results and just comment on the sign and significance of different factors, showing that previous papers have already found such results. However, no clear implication of these results are provided. On the other hand, results lack any causal interpretation since they are obviously plagued by endogeneity and self-selection problems (eg., labour market participation), Therefore, these correlation results are barely useful for policy interventions as the authors claim. For example, imagine that job opportunities in urban areas are lower for woman, while in rural areas some opportunities always exist. Then, most women in urban areas decide NOT to join to the labour force (they are not “currently available for work” and “seeking work”), but almost all of the few ones that join find a job (fill those few job opportunities); therefore, unemployment is low ( (“without work” AMONG those in the labour force: “currently available for work” and “seeking work”). The policy recommendation by authors is to focus on rural areas where the problem exists, while job opportunities are equally or less scarce in urban areas.

In summary, this work has a number of important shortcomings and its contribution to the literature is unclear. I do not feel that it reaches the standards of a scientific publication in an international journal.

6. PLOS authors have the option to publish the peer review history of their article (what does this mean?). If published, this will include your full peer review and any attached files.

Reviewer #1: No

Reviewer #2: No

---

## [Author Response · Author response to Decision Letter 0]

21 Jan 2022

First of all thanks for your worth and essential over all comments. On the very beginning I am glad for your nice reviewing and assessing this article, entitled, Geographical variation and determinants of women unemployment status in Ethiopia; A multilevel and spatial analysis from 2016 Ethiopia Demographic and Health Survey data.

Editor Comments :

Comment 1:Please ensure that your manuscript meets PLOS ONE's style requirements, including those for file naming. The PLOS ONE style templates can be found at 

Response: Thank you for your important comments. We have followed all PLOS ONE's style requirements, including those for file naming.

Comment 2:Please amend either the abstract on the online submission form (via Edit Submission) or the abstract in the manuscript so that they are identical.

Response: Thank you for your suggestion. We correct the abstract in our manuscript to be identical.

Comment 3: Your ethics statement should only appear in the Methods section of your manuscript. If your ethics statement is written in any section besides the Methods, please move it to the Methods section and delete it from any other section. Please ensure that your ethics statement is included in your manuscript, as the ethics statement entered into the online submission form will not be published alongside your manuscript. 

Response: Thank you for your suggestion.we have incorporated the points that the ethical statement is only appear in the method section.

Comment 4: We note that Figures 4.1,4.2 and 4.3 in your submission contain map images which may be copyrighted. All PLOS content is published under the Creative Commons Attribution License (CC BY 4.0), which means that the manuscript, images, and Supporting Information files will be freely available online, and any third party is permitted to access, download, copy, distribute, and use these materials in any way, even commercially, with proper attribution. For these reasons, we cannot publish previously copyrighted maps or satellite images created using proprietary data, such as Google software (Google Maps, Street View, and Earth). For more information, see our copyright guidelines: http://journals.plos.org/plosone/s/licenses-and-copyright.

A. You may seek permission from the original copyright holder of Figures 4.1,4.2 and 4.3 to publish the content specifically under the CC BY 4.0 license. 

B. If you are unable to obtain permission from the original copyright holder to publish these figures under the CC BY 4.0 license or if the copyright holder’s requirements are incompatible with the CC BY 4.0 license, please either i) remove the figure or ii) supply a replacement figure that complies with the CC BY 4.0 license. Please check copyright information on all replacement figures and update the figure caption with source information. If applicable, please specify in the figure caption text when a figure is similar but not identical to the original image and is therefore for illustrative purposes only.

Response: Thank you for your suggestion. We incorporate your worth comments and the map images are not copyrighted, which are my work.

Comment 5: Thank you for submitting the above manuscript to PLOS ONE. During our internal evaluation of the manuscript, we found significant text overlap between your submission and the following previously published works.

- https://bmcpublichealth.biomedcentral.com/articles/10.1186/s12889-019-7529-z

- https://bmjopen.bmj.com/content/9/4/e027276

- https://pubmed.ncbi.nlm.nih.gov/24411023/

Please revise the manuscript to rephrase the duplicated text, cite your sources, and provide details as to how the current manuscript advances on previous work. Please note that further consideration is dependent on the submission of a manuscript that addresses these concerns about the overlap in text with published work.

Response: Thank you for your suggestion. We incorporate your worth comments and we revise and paraphrase all the manuscript in detail.

Reviewer #1: 

Comment 1:I am glad to review and assess this interesting article, entitled, Geographical variation and determinants of women unemployment status in Ethiopia; A multilevel and spatial analysis from 2016 Ethiopia Demographic and Health Survey data. The organization of this article is good and satisfactory. The Introduction part and methodology portions are adequate. I suggest the authors improve the Introduction section by adding some latest articles' citations to enhance the work quality.

Overall, the manuscript is a good piece of work. I recommend that authors do a little more work and add the latest literature to support the study, as suggested. The English level is not good and smooth, e.g., the language standard, specifically the grammar, of not sufficient quality to meet scientific merit for publication. I accept this manuscript after minor revision, as I have recommended.

Response: Thank you for your suggestion and comments. This is a very essential comment, we have incorporated the points. We repeatedly read the whole document and consider all the corrections (adding the most recent citation and also incorporate the latest literature) and the language is edited by professionals.

Reviewer #2:

Comment1:This work contains two simple empirical exercises about female unemployment in Ethiopia.

In the first one, the authors routinely use standard spatial techniques to detect clusters of areas with low or high unemployment. While technically well carried out (mainly because it is simply implemented in usual software), results are barely commented and it is particularly unclear the contribution of this analysis. The potential contribution should be focused on commenting about spatial correlation. To start with, both Figure 4.2 and 4.3 are barely commented. The terms used in Figure 4.2 are confusing: "High Cluster" can also apply to what the authors call "High outlier", because (1) both are clusters (but in the latter case this is not mentioned), (2) the term outlier is not appropriate because being a high-low area is NOT weird (=outlier), but a simple indication of negative spatial correlation. If the authors could just READ their software output (labels in the picture their software provided), they could find the standard term: high-high cluster and high-low cluster. In any case, authors just provide a purely descriptive minimum comments on the areas with high or low unemployment incidence, but no discussion at all about the implications or how these results can be useful. Commenting that female unemployment is different in different clusters is very vague and can be concluded from any other study. And the general recommendations in the last two sentences of the Conclusions are NOT directly supported by the analysis.

Therefore, it is unclear the value added by the spatial analysis. The authors do not seem to feel that it was useful, because the second part of their work does not use the information about spatial correlation among clusters (enumeration areas). They just apply a multilevel logistic model, but they do not seem to take into account spatial correlation to compute standard errors (and therefore confidence interval). It is, of course, important to account for both individual level and community/clusters level factors to avoid biased results (i.e., as the authors mention, incorrectly concluding that a factor is significant or not). But the spatial correlation analysis is not needed to account for this. If one finds that spatial correlation exists, this should be incorporated into the subsequent analysis. In their current analysis, the authors allow for within-cluster (community) correlation, but assume no between-clusters correlation despite of their previous findings!

The (logistic) regression analysis is just a sophisticated but ultimately simple exercise about which variables are correlated with observed female unemployment. On the one hand, no really new result is found. The authors again present the results and just comment on the sign and significance of different factors, showing that previous papers have already found such results. However, no clear implication of these results are provided. On the other hand, results lack any causal interpretation since they are obviously plagued by endogeneity and self-selection problems (eg., labour market participation), Therefore, these correlation results are barely useful for policy interventions as the authors claim. For example, imagine that job opportunities in urban areas are lower for woman, while in rural areas some opportunities always exist. Then, most women in urban areas decide NOT to join to the labour force (they are not “currently available for work” and “seeking work”), but almost all of the few ones that join find a job (fill those few job opportunities); therefore, unemployment is low ( (“without work” AMONG those in the labour force: “currently available for work” and “seeking work”). The policy recommendation by authors is to focus on rural areas where the problem exists, while job opportunities are equally or less scarce in urban areas.

In summary, this work has a number of important shortcomings and its contribution to the literature is unclear. I do not feel that it reaches the standards of a scientific publication in an international journal.

Response: On the very beginning I am glad for your nice reviewing and assessing this article, entitled, Geographical variation and determinants of women unemployment status in Ethiopia; A multilevel and spatial analysis from 2016 Ethiopia Demographic and Health Survey data.

Then for your question on fig4.2 and fig4.3, I need to clarify that, initially the research routinely used to show the geographical variation/distribution of unemployment rate of women by using spatial auto-correlation. So, the Global Moran’s I Spatial auto-correlation indices measure the spatial dependence between values of the same variable(unemployment rate) in different places in space. Then we say that, Spatial auto-correlation is positive when similar values(high unemployment rate with high unemployment rate or low unemployment rate with low unemployment rate) of the variable to be studied are grouped geographically.Spatial auto-correlation is negative when the dissimilar values of the variable(high low or low high) to be studied come together geographically.

In other words, spatial dependence exists when statistical values are correlated. However, in our case Global Moran’s I values I=0.33 (p-value=0.001) , this can only explain the clustering(non-random) effects in a given enumeration areas without clearly highlighting the clustering regions. This method was most commonly used in testing global spatial auto-correlation. Often, our interest lies not only in determining whether the data as a whole exhibit spatial auto-correlation, but also, in identifying the specific observations that exhibit spatial auto-correlation with their neighbors.So,the study following local measures called local spatial auto-correlation indicators(LISA), which is a set of local indicators for inferring the scope of clustering regions (in our case enumeration areas). Once a significance level is set(if the global spatial auto-correlation is significant) , values can also be plotted on a map to display the specific locations of hot spots and potential outliers(dissimilar values). The results of LISA map are displayed on fig4.2 and fig4.3. Statistical significance tests are mainly performed to test the scope of clustering spatial elements relative to the entire scope, where a higher significance denotes that spatial clustering is more prominent. 

However, in fig4.2 result we try to clarify the potential outliers areas on unemployment rate of women as:high outlier(high-low) refers to high proportion of women unemployment surrounded by low proportion of women unemployment.; low outlier(low-high)refers to low proportion of women unemployment surrounded by high proportion of women unemployment. And the rest was similar values.

And then, in fig4.3 ahot spot(high risk areas) and cold spot(low risk areas) analysis of the risk areas for women'sunemployment rate in Ethiopia has been summarized.

After all the spatial analysis result were discussed on the first paragraph in discussion part and concluded from any other study.

After all, concluding that the spatial distribution of women's unemployment status is non-random( I=0.33 (p-value=0.001)) at the national level and associated with neighboring values; and the data set taken for this study was from EDHS 2016. These data are hierarchical structure and surveys are obtained from nested sampling in heterogeneous subgroups or a sampling method was multistage stratified cluster sampling. For multistage clustered samples, the dependence among observations often comes from several levels of the hierarchy. The problem of dependencies between individual observations also occurs in survey research, where the sample is not taken randomly but cluster sampling from geographical areas is used instead. In this case, the use of single-level statistical models is no longer valid and reasonable. Hence, in order to draw appropriate inferences and conclusions from multistage stratified clustered survey data we may require tricky and complicated modeling techniques like multilevel modeling. That is why we use multilevel logistic regression model by take into account for lack of independence(correlation) across levels of nested data (i.e., individuals (women) nested within enumeration areas). This information also supported by the intra-class correlation coefficient (ICC) measures the proportion of variance in the outcome explained by the grouping structure. This ICC is an indication of the correlation of the women unemployment rate belonging to the same enumeration areas, i.e. it is an indication of the dependency of the unemployment rate among women within the enumeration areas. So,from the research analysis, About 21.48% of the variation in women's unemployment rate occurred due to community-level factors (ICC =21.48%).The MOR (2.46) value of women's unemployment was the largest in the null model, which showed that there was variation between communities clustered as MOR was 2.46 times higher than the comparison (MOR=1.78). In addition, the highest (58.89%) PCV in the full model (model IV) revealed that about 59%of the difference in women's unemployment across populations was due to both the individual and community-level factors. The unexplained community variation in women unemployment decreased to MOR of 1.78 when all factors were added to the null model (empty model). This indicates that when all factors are included, the effect of clustering is still statistically significant in the full model (Table 3).

---

## [Decision Letter · Decision Letter 1]

25 Mar 2022

PONE-D-21-08056R1Geographical variation and determinants of women unemployment status in Ethiopia; A multilevel and spatial analysis from 2016 Ethiopia Demographic and Health Survey dataPLOS ONE

Dear Dr. Fenta,

Thank you for submitting your manuscript to PLOS ONE. After careful consideration, we feel that it has merit but does not fully meet PLOS ONE’s publication criteria as it currently stands. Therefore, we invite you to submit a revised version of the manuscript that addresses the points raised during the review process.

Please note the comments raised by the reviewers below. In particular, please address the comments from Reviewer #2 regarding outliers and standard errors.

We look forward to receiving your revised manuscript.

Kind regards,

Hugh Cowley

Senior Editor

PLOS ONE

Reviewers' comments:

Reviewer's Responses to Questions

**Comments to the Author**

1. If the authors have adequately addressed your comments raised in a previous round of review and you feel that this manuscript is now acceptable for publication, you may indicate that here to bypass the “Comments to the Author” section, enter your conflict of interest statement in the “Confidential to Editor” section, and submit your "Accept" recommendation.

Reviewer #1: All comments have been addressed

Reviewer #2: (No Response)

2. Is the manuscript technically sound, and do the data support the conclusions?

Reviewer #1: Yes

Reviewer #2: Partly

3. Has the statistical analysis been performed appropriately and rigorously? 

Reviewer #1: Yes

Reviewer #2: No

4. Have the authors made all data underlying the findings in their manuscript fully available?

Reviewer #1: Yes

Reviewer #2: Yes

5. Is the manuscript presented in an intelligible fashion and written in standard English?

Reviewer #1: Yes

Reviewer #2: Yes

6. Review Comments to the Author

Reviewer #1: I am glad to review this exciting article . I directly recommend this study for publication . Fully satisfied

Reviewer #2: I appreciate that the authors have tried to clarify some of my previous comments. But my comments have not addressed no proper argument has been provided to justify their choices.

I do not see any contribution in finding spatial correlation in unemployment: I doubt that in any country one can find that unemployment is randomly distributed across regions (which are endogenously formed in a way clearly linked to unemployment). The authors' response has ignored my point about the incorrect use of the word "outliers" as the authors do: this is not a standard way to name that situation, a better one exists and it is confusing.

I do not see again that my comments on the second part of the paper (factors associated with women unemployment) have been addressed. As I said, this provides some mere correlations between factors and unemployment. This descriptive evidence can have some interest (this is an editorial choice), but it should be properly carried out AND it must be clear that NO CAUSAL interpretation should be concluded from it. On the one hand, if the authors carefully read my previous previous, I mention that STANDARD ERRORS should be account for clustering; I did not mention at all, as the authors focus on their reply, on using single-level or multilevel statistical models. If standard errors are not properly computed, results are not credible. On the other hand, the authors should be much clearer about the merely descriptive implications of their results. They cannot claim (as they implicitly and explicitly do) that a changing some factors (eg., improving entrepreneurship,etc.) could have an effect on unemployment: this a causal claim that cannot be derived from the analysis and this should be clear in a serious scientific paper.

7. PLOS authors have the option to publish the peer review history of their article (what does this mean?). If published, this will include your full peer review and any attached files.

Reviewer #1: No

Reviewer #2: No

---

## [Author Response · Author response to Decision Letter 1]

8 Apr 2022

First of all thanks for your worth and essential over all comments. On the very beginning I am glad for your nice reviewing and assessing this article, entitled, Geographical variation and determinants of women unemployment status in Ethiopia; A multilevel and spatial analysis from 2016 Ethiopia Demographic and Health Survey data.

Reviewer #1:I am glad to review this exciting article . I directly recommend this study for publication . Fully satisfied 

Response: We authors owe you a great debt of gratitude.

Reviewer #2: I appreciate that the authors have tried to clarify some of my previous comments. But my comments have not addressed no proper argument has been provided to justify their choices. I do not see any contribution in finding spatial correlation in unemployment: I doubt that in any country one can find that unemployment is randomly distributed across regions (which are endogenously formed in a way clearly linked to unemployment). The authors' response has ignored my point about the incorrect use of the word "outliers" as the authors do: this is not a standard way to name that situation, a better one exists and it is confusing. I do not see again that my comments on the second part of the paper (factors associated with women unemployment) have been addressed. As I said, this provides some mere correlations between factors and unemployment. This descriptive evidence can have some interest (this is an editorial choice), but it should be properly carried out AND it must be clear that NO CAUSAL interpretation should be concluded from it. On the one hand, if the authors carefully read my previous previous, I mention that STANDARD ERRORS should be account for clustering; I did not mention at all, as the authors focus on their reply, on using single-level or multilevel statistical models. If standard errors are not properly computed, results are not credible. On the other hand, the authors should be much clearer about the merely descriptive implications of their results. They cannot claim (as they implicitly and explicitly do) that a changing some factors (eg., improving entrepreneurship,etc.) could have an effect on unemployment: this a causal claim that cannot be derived from the analysis and this should be clear in a serious scientific paper.

Response: At the outset, we want to thank you for taking the time to review and evaluate this article again.

The research is routinely used to demonstrate the initial goal of investigating the spatial structure of unemployment rate across different clusters to provide implications for policymakers, investigating the hot spots of unemployment rate, and showing a visual picture of unemployment rate. Mapping was used as a preliminary step in conducting a visual inspection for the unemployment rate. In addition, mapping is important in the monitoring of unemployed people. Maps can reveal spatial patterns that were previously unknown or unnoticed when examining a table of statistics, as well as high-risk communities or problem areas. The goal of spatial analysis is to find patterns in geographic data and try to explain them. Spatial autocorrelation is the term used for theinterdependence of the values of a variable over space.As a result, the most important contribution in determining spatial correlation/statistics in unemployment is not only interested in answering the "how much unemployment rate women in Ethiopia" question, but also the "how much is where" question. As stated in the first law of geography, "everything is related to everything else, but closer things are more related than distant things," the application of statistical techniques to spatial data faces significant challenges. Two spatial autocorrelation statistics based on sharing boundary neighbors, known as global and local Moran's I, were used to investigate global clustering and local clusters, respectively. Based on visual inspection of the mapping, global clustering was discovered in unemployment rate, which was confirmed by the significant statistic discovered by global Moran's I(Moran’s I values I=0.33 (p-value=0.001).

Our interest is frequently not only in determining whether the data as a whole exhibits spatial auto-correlation, but also in identifying specific observations that exhibit spatial auto-correlation with their neighbors. As a result, the study relied on local measures known as local spatial auto-correlation indicators (LISA), which are a collection of local indicators used to determine the scope of clustering regions (in our case enumeration areas). Once a significance level has been determined (if the global spatial auto-correlation is significant), values can be plotted on a map to show the precise locations of hot spots and potential outliers (dissimilar values) or A positive local Moran value indicates local stability, such as a cluster with a high/low unemployment rate surrounded by another cluster with a high/low unemployment rate. A negative local Moran value indicates local instability(high/low outlier ), such as a cluster with low unemployment surrounded by a cluster with high unemployment, or vice versa. Figures 2 and 3 show the results of the LISA map.

And ,Thank you for your suggestion and comments. This is a very essential comment, we have incorporated the points. We repeatedly read the whole document and consider all the corrections.

After all, concluding that the spatial distribution of women's unemployment status at the national level is non-random (I=0.33; p-value=0.001) and associated with neighboring values. The result indicates that there is spatial autocorrelation /dependency or the unemployment rate is interdependent across clusters. As a result, we employ a multilevel logistic regression model that accounts for a lack of independence (spatial correlation) across levels of nested data (i.e., individuals (women) nested within clusters). The spatial correlation result was also supported by the intra-class correlation coefficient (ICC), which measures the proportion of variance in the outcome explained by the grouping structure while accounting for variance across clusters. This ICC represents the correlation of the women's unemployment rate within the same enumeration areas, i.e. the dependency of the unemployment rate among women within the clusters.So,from the research analysis, About 21.48% of the variation in women's unemployment rate occurred due to community-level(due to clusters) factors (ICC =21.48%).

---

## [Decision Letter · Decision Letter 2]

29 Apr 2022

PONE-D-21-08056R2Geographical variation and determinants of women unemployment status in Ethiopia; A multilevel and spatial analysis from 2016 Ethiopia Demographic and Health Survey dataPLOS ONE

Dear Dr. Setegn Muche Fenta,

Thank you for submitting your manuscript to PLOS ONE. After careful consideration, we feel that it has merit but does not fully meet PLOS ONE’s publication criteria as it currently stands. Therefore, we invite you to submit a revised version of the manuscript that addresses the points raised during the review process.

The reviewer has some major concerns about the revision. Please provide more evidence or more detail discuss in the next version. Please submit your revised manuscript by June 13, 2022. If you will need more time than this to complete your revisions, please reply to this message or contact the journal office at plosone@plos.org. Please include the following items when submitting your revised manuscript:A rebuttal letter that responds to each point raised by the academic editor and reviewer(s). You should upload this letter as a separate file labeled 'Response to Reviewers'.A marked-up copy of your manuscript that highlights changes made to the original version. You should upload this as a separate file labeled 'Revised Manuscript with Track Changes'.An unmarked version of your revised paper without tracked changes. You should upload this as a separate file labeled 'Manuscript'.

We look forward to receiving your revised manuscript.

Kind regards,

Wen-Wei Sung, M.D., Ph.D.

Academic Editor

PLOS ONE

Reviewers' comments:

Reviewer's Responses to Questions

**Comments to the Author**

1. If the authors have adequately addressed your comments raised in a previous round of review and you feel that this manuscript is now acceptable for publication, you may indicate that here to bypass the “Comments to the Author” section, enter your conflict of interest statement in the “Confidential to Editor” section, and submit your "Accept" recommendation.

Reviewer #2: (No Response)

2. Is the manuscript technically sound, and do the data support the conclusions?

Reviewer #2: Partly

3. Has the statistical analysis been performed appropriately and rigorously? 

Reviewer #2: No

4. Have the authors made all data underlying the findings in their manuscript fully available?

Reviewer #2: Yes

5. Is the manuscript presented in an intelligible fashion and written in standard English?

Reviewer #2: Yes

6. Review Comments to the Author

Reviewer #2: I do not see any real improvement with respect to the previous versions. None of my comments that the editor mentioned have been properly addressed.

I feel that a crucial point in any serious scientific paper as making clear that the results cannot be interpreted in a causal way CANNOT be addressed just by adding a "could be". Instead it should be explicitly mentioned the true purely descriptive scope of the results.

The authors do not only insist in using the term "outlier" in a unconventional sense, but they have now extended its use. Even more surprisingly they do not even attempt to provide a simple explanation about their choice, although I have mentioned this in my two previous reports. It is really a complete outlier the way in which the author insist in using the term "outlier" (positive correlation between two variables, say, high education and high income is never called a positive outlier). And the way in which the authors completely ignore this point (to accept it or to provide a convincing alternative explanation about their unusual way of using the term) is more outlier in peer review process.

I perfectly know that the interesting question is to show where the unemployment is located and how is related to nearby areas. I never raised this point that the authors insist in explaining while ignoring other more important and explicitly mentioned. However, the paper is more focused precisely in trivial and uninteresting things: to discussed the global Moran statistics that reveals the TRIVIAL point that unemployment is not uniform across regions (randomly distributed) as I already mentioned.

I perfectly know how to interpret the local spatial auto-correlation indicators. If the authors carefully read and understand this and my previous reports, they would see that the literature in this area NEVER calls OUTLIERS to what is simply positive or negative correlation amongst clusters (high/high, high/low, etc.). The term outliers is NOT used in this or any other literature to name dissimilar values (high/low); and, as I already mention, the standard output of statistical software already provides a proper term for this (and it is not outlier). Even more, if the authors were using outlier in the sense of dissimilar values (which is unconventional and not justified or explained by the authors), high-high and low-low cases are also identified as "outliers" but in this case the values are not dissimilar...

Finally the authors have completely ignored (again) my point about standard errors. Statistical modelling (i.e., using a multilevel model or not) is not the only relevant part of empirical results; and in any case it was never my point. In any empirical papers, researcher should be very careful about inference. And in this case it means to use proper standard errors. Otherwise, the results could be meaningless: some factors that the authors claim to be correlated to unemployment might actually be not significantly related to it. Once again the authors do not even mention why this crucial point is not even properly addressed.

7. PLOS authors have the option to publish the peer review history of their article (what does this mean?). If published, this will include your full peer review and any attached files.

Reviewer #2: No

---

## [Author Response · Author response to Decision Letter 2]

17 May 2022

Response to Reviewers

First of all thanks for your worth and essential over all comments. On the very beginning I am glad for your nice reviewing and assessing this article, entitled, Geographical variation and determinants of women unemployment status in Ethiopia; A multilevel and spatial analysis from 2016 Ethiopia Demographic and Health Survey data.

Reviewer #2: I do not see any real improvement with respect to the previous versions. None of my comments that the editor mentioned have been properly addressed.

I feel that a crucial point in any serious scientific paper as making clear that the results cannot be interpreted in a causal way CANNOT be addressed just by adding a "could be". Instead it should be explicitly mentioned the true purely descriptive scope of the results.

The authors do not only insist in using the term "outlier" in a unconventional sense, but they have now extended its use. Even more surprisingly they do not even attempt to provide a simple explanation about their choice, although I have mentioned this in my two previous reports. It is really a complete outlier the way in which the author insist in using the term "outlier" (positive correlation between two variables, say, high education and high income is never called a positive outlier). And the way in which the authors completely ignore this point (to accept it or to provide a convincing alternative explanation about their unusual way of using the term) is more outlier in peer review process.

I perfectly know that the interesting question is to show where the unemployment is located and how is related to nearby areas. I never raised this point that the authors insist in explaining while ignoring other more important and explicitly mentioned. However, the paper is more focused precisely in trivial and uninteresting things: to discussed the global Moran statistics that reveals the TRIVIAL point that unemployment is not uniform across regions (randomly distributed) as I already mentioned.

I perfectly know how to interpret the local spatial auto-correlation indicators. If the authors carefully read and understand this and my previous reports, they would see that the literature in this area NEVER calls OUTLIERS to what is simply positive or negative correlation amongst clusters (high/high, high/low, etc.). The term outliers is NOT used in this or any other literature to name dissimilar values (high/low); and, as I already mention, the standard output of statistical software already provides a proper term for this (and it is not outlier). Even more, if the authors were using outlier in the sense of dissimilar values (which is unconventional and not justified or explained by the authors), high-high and low-low cases are also identified as "outliers" but in this case the values are not dissimilar...

Finally the authors have completely ignored (again) my point about standard errors. Statistical modelling (i.e., using a multilevel model or not) is not the only relevant part of empirical results; and in any case it was never my point. In any empirical papers, researcher should be very careful about inference. And in this case it means to use proper standard errors. Otherwise, the results could be meaningless: some factors that the authors claim to be correlated to unemployment might actually be not significantly related to it. Once again the authors do not even mention why this crucial point is not even properly addressed..

Response: And ,Thank you for your suggestion and comments. This is a very essential comment, we have incorporated the points. We repeatedly read the whole document and consider all the corrections.

At the outset, we want to thank you for taking the time to review and evaluate this article again. The manuscript was revised (all the spatial out put was updated).

Your comment is clear but please understated the aim of this study.

This research is routinely used to assess the spatial distribution (varation) of unemployment rate across different clusters to provide implications for policymakers, investigating the hot spots of unemployment rate(the highest risk region/areas), and showing a visual picture of unemployment rate in the first section.

 In this part we identify that:

Spatial pattern of women unemployment

In Ethiopia, the spatial distribution of women's unemploymentwas spatially clustered with Global Moran's I = 0.064298 (z-score = 16.375386, P-value <0.0001). This demonstrated that spatial hotspot and coldspot clustering was identified in Ethiopian regions. Given the z-score of 16.375386, there was less than a 1% chance that this high-clustered pattern was the result of randomchance. The bright red and blue colors on the tails indicate a higher level of significance (Fig. 2).

Then we find the hot spot also:

Hot spot (Getis-Ord Gi*) analysis

Figure3 summarizes a hot and cold spot analysis of the risk locations for women's unemployment in Ethiopia. Beneshagul and Gambela, Oromia, WesternAmhara, and northeren SNNPR are the regions with the highest unemployment rates. Tigray, Afar, Dire Dawa, Hariri, northern Amhara, and Somalia on the other hand, have been designated as cold-spot (low risk) regions (Fig 3).

And we can identify the most likely cluster by Spatial scan statistics:

Spatial SaTScan analysis of unemployment women across region

Most likely (primary clusters) and secondary clusters of unemployment women were identified. In 2016 EDHS, spatial scan statistics identified a total of high and modest performing spatial clusters of unemployment women. A total of 164 significant clusters were identified. Of these, among the significant clusters, 120 were most likely (primary cluster) and the other 44 were secondary.The spatial window of the primary cluster was located at Tigray, Amhara, southern Gambela,eastern Oromia and Afar which was centered at (12.376936 N,38.357984 E) / 318.07 km, RR = 1.43, and log likelihood ratio (LLR) of 200.78 at p-value < 0.0001. It showed that house holds in the spatial window had 1.43 times higher unemployment women ratethan those outside the window (Table 2 and Fig. 4).

 Here we are addressing the spatial distribution and the risk area/cluster of unemployment, which means we speak about the first aim of this study.

In the next section of this study: We need to identify the determinant factors of women unemployment rate in Ethiopia

After all, concluding that the spatial distribution of women's unemployment status at the national level is non-random and associated with neighboring values. The result indicates that there is spatial autocorrelation /dependency or the unemployment rate is interdependent across clusters. As a result, we employ a multilevel logistic regression model that accounts for a lack of independence (spatial correlation) across levels of nested data (i.e., individuals (women) nested within clusters). The spatial correlation result was also supported by the intra-class correlation coefficient (ICC), which measures the proportion of variance(indirectly we know the standard deviation) in the outcome explained by the grouping structure while accounting for variance across clusters. This ICC represents the correlation of the women's unemployment rate within the same enumeration areas, i.e. the dependency of the unemployment rate among women within the clusters.So,from the research analysis, About 21.48% of the variation in women's unemployment rate occurred due to community-level(due to clusters) factors (ICC =21.48%).and MOR is showing that there was variation between communities clustered.So, here we know that the within and between cluster variation. 

The result and interpretation was clearly mentioned in the manuscript. 

Next, we examined the factors that influenced Ethiopia's unemployment rate using MLM, since this model incorporates the cluster correlation between unemployment and various factors.

---

## [Decision Letter · Decision Letter 3]

13 Jun 2022

PONE-D-21-08056R3Geographical variation and determinants of women unemployment status in Ethiopia; A multilevel and spatial analysis from 2016 Ethiopia Demographic and Health Survey dataPLOS ONE

Dear Dr. Setegn Muche Fenta,

Thank you for submitting your manuscript to PLOS ONE. After careful consideration, we feel that it has merit but does not fully meet PLOS ONE’s publication criteria as it currently stands. Therefore, we invite you to submit a revised version of the manuscript that addresses the points raised during the review process.

We look forward to receiving your revised manuscript.

Kind regards,

Wen-Wei Sung, M.D., Ph.D.

Academic Editor

PLOS ONE

Journal Requirements:

Reviewers' comments:

Reviewer's Responses to Questions

**Comments to the Author**

1. If the authors have adequately addressed your comments raised in a previous round of review and you feel that this manuscript is now acceptable for publication, you may indicate that here to bypass the “Comments to the Author” section, enter your conflict of interest statement in the “Confidential to Editor” section, and submit your "Accept" recommendation.

Reviewer #2: (No Response)

2. Is the manuscript technically sound, and do the data support the conclusions?

Reviewer #2: Partly

3. Has the statistical analysis been performed appropriately and rigorously? 

Reviewer #2: Yes

4. Have the authors made all data underlying the findings in their manuscript fully available?

Reviewer #2: Yes

5. Is the manuscript presented in an intelligible fashion and written in standard English?

Reviewer #2: Yes

6. Review Comments to the Author

Reviewer #2: I do not see how my points were at odds with the aim of the study. Identifying hot spots areas is fine, but one cannot use concepts (i.e., outlier) in a completely different way as the literature does or one must provide good arguments for such unconventional choice. The authors have gone much beyond this modest point and now provide a completely new set of results. While I did not ask to follow this approach, I acknowledge that it has improved a lot the text and I prefer these results to the previous ones even if they had included my proposed corrections. The current results are in the line with the previous ones, although they are more complete and provide better insights.

In the previous version, the authors defined/used "outlier" in an incorrect way according to the literature. Now the term "outlier" is only used a couple of times in the Discussion section, but never properly defined. An "outlier" does not mean any value just above or below the mean or a reference value, but a really extreme value. It is never clear in the main text how an outlier is defined; for instance, how many times apart from mean/reference value and WHY, as it is routinely done in any rigorous statistical publication dealing with outliers.

7. PLOS authors have the option to publish the peer review history of their article (what does this mean?). If published, this will include your full peer review and any attached files.

Reviewer #2: No

---

## [Author Response · Author response to Decision Letter 3]

16 Jun 2022

Thank you for your comments

Journal Requirements:

Please review your reference list to ensure that it is complete and correct. If you have cited

papers that have been retracted, please include the rationale for doing so in the manuscript text,

or remove these references and replace them with relevant current references. Any changes to

the reference list should be mentioned in the rebuttal letter that accompanies your revised

manuscript. If you need to cite a retracted article, indicate the article’s retracted status in the

References list and also include a citation and full reference for the retraction notice.

Response: Please accept our sincere thanks for your comment. We carefully review our reference list and ensure that it is accurate. Additional references are also added to the list.

31. Broussard, N. and T.G. Tekleselassie, Youth unemployment: Ethiopia country study. International Growth Centre. Working Paper, 2012. 12(0592): p. 1-37.

32. Nganwa, P., D. Assefa, and P. Mbaka, The nature and determinants of urban youth unemployment in Ethiopia. Nature, 2015. 5(3): p. 197-203.

33. Batu, M.M., Determinants of youth unemployment in urban areas of Ethiopia. International Journal of Scientific and Research Publications, 2016. 6(5): p. 343-350.

Reviewer #2:

 I do not see how my points were at odds with the aim of the study. Identifying hot spots areas is fine, but one cannot use concepts (i.e., outlier) in a completely different way as the literature does or one must provide good arguments for such unconventional choice. The authors have gone much beyond this modest point and now provide a completely new set of results. While I did not ask to follow this approach, I acknowledge that it has improved a lot the text and I prefer these results to the previous ones even if they had included my proposed corrections. The current results are in the line with the previous ones, although they are more complete and provide better insights.

In the previous version, the authors defined/used "outlier" in an incorrect way according to the literature. Now the term "outlier" is only used a couple of times in the Discussion section, but never properly defined. An "outlier" does not mean any value just above or below the mean or a reference value, but a really extreme value. It is never clear in the main text how an outlier is defined; for instance, how many times apart from mean/reference value and WHY, as it is routinely done in any rigorous statistical publication dealing with outliers.

Response: Please accept our sincere thanks for your comment. All your comments have been incorporated into the current version of the manuscript.

---

## [Editor Report · Decision Letter 4]

22 Jun 2022

Geographical variation and determinants of women unemployment status in Ethiopia; A multilevel and spatial analysis from 2016 Ethiopia Demographic and Health Survey data

PONE-D-21-08056R4

Dear Dr. Setegn Muche Fenta,

We’re pleased to inform you that your manuscript has been judged scientifically suitable for publication and will be formally accepted for publication once it meets all outstanding technical requirements.

Kind regards,

Wen-Wei Sung, M.D., Ph.D.

Academic Editor

PLOS ONE
---

## [Editor Report · Acceptance letter]

28 Jun 2022

PONE-D-21-08056R4 

Geographical variation and determinants of women unemployment status in Ethiopia; A multilevel and spatial analysis from 2016 Ethiopia Demographic and Health Survey data 

Dear Dr. Fenta:

I'm pleased to inform you that your manuscript has been deemed suitable for publication in PLOS ONE. Congratulations! Your manuscript is now with our production department. 

Kind regards, 

on behalf of

Dr. Wen-Wei Sung 

Academic Editor

PLOS ONE